# A Unified Model for Multi-class Anomaly Detection

**Zhiyuan You**[1*] **Lei Cui**[2*] **Yujun Shen**[3] **Kai Yang**[4] **Xin Lu**[4] **Yu Zheng**[1] **Xinyi Le**[1†]

[1]Shanghai Jiao Tong University    [2]Tsinghua University    [3]CUHK    [4]SenseTime

zhiyuanyou@foxmail.com, cuil19@mails.tsinghua.edu.cn, shenyujun0302@gmail.com
{yangkai, luxin}@sensetime.com, yuzheng@sjtu.edu.cn, lexinyi@sjtu.edu.cn

## Abstract

Despite the rapid advance of unsupervised anomaly detection, existing methods require to train separate models for different objects. In this work, we present *UniAD* that accomplishes anomaly detection for multiple classes with a unified framework. Under such a challenging setting, popular reconstruction networks may fall into an "identical shortcut", where both normal and anomalous samples can be well recovered, and hence fail to spot outliers. To tackle this obstacle, we make three improvements. First, we revisit the formulations of fully-connected layer, convolutional layer, as well as attention layer, and confirm the important role of query embedding (*i.e.*, within attention layer) in preventing the network from learning the shortcut. We therefore come up with a *layer-wise query decoder* to help model the multi-class distribution. Second, we employ a *neighbor masked attention* module to further avoid the information leak from the input feature to the reconstructed output feature. Third, we propose a *feature jittering* strategy that urges the model to recover the correct message even with noisy inputs. We evaluate our algorithm on MVTec-AD and CIFAR-10 datasets, where we surpass the state-of-the-art alternatives by a sufficiently large margin. For example, when learning a unified model for 15 categories in MVTec-AD, we surpass the second competitor on the tasks of both anomaly detection (from 88.1% to 96.5%) and anomaly localization (from 89.5% to 96.8%). Code is available at https://github.com/zhiyuanyou/UniAD.

## 1 Introduction

Anomaly detection has found an increasingly wide utilization in manufacturing defect detection [4], medical image analysis [17], and video surveillance [46]. Considering the highly diverse anomaly types, a common solution is to model the distribution of normal samples and then identify anomalous ones via finding outliers. It is therefore crucial to learn a compact boundary for normal data, as shown in Fig. 1a. For this purpose, existing methods [6, 11, 25, 27, 48, 49, 52] propose to train separate models for different classes of objects, like in Fig. 1c. However, such a one-class-one-model scheme could be memory-consuming especially along with the number of classes increasing, and also uncongenial to the scenarios where the normal samples manifest themselves in a large intra-class diversity (*i.e.*, one object consists of various types).

In this work, we target a more practical task, which is to detect anomalies from different object classes with a unified framework. The task setting is illustrated in Fig. 1d, where the training data covers normal samples from a range of categories, and the learned model is asked to accomplish anomaly detection for all these categories without any fine-tuning. It is noteworthy that the categorical information (*i.e.*, class label) is inaccessible at both the training and the inference stages, considerably

---

[*] Contribute Equally.    [†] Corresponding Author.

36th Conference on Neural Information Processing Systems (NeurIPS 2022).

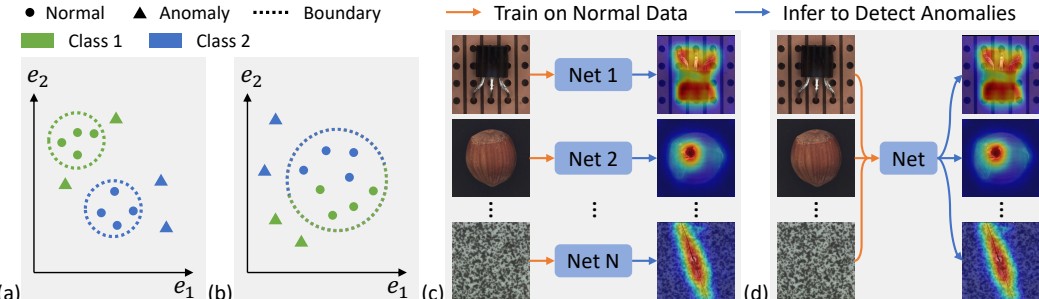

Figure 1: **Task setting of unified anomaly detection.** (a) Existing methods learn separate decision boundaries for different object classes, while (b) our approach models the multi-class data distribution such that one boundary is enough to spot outliers regarding all categories. As a result, we escape from the conventional one-class-one-model paradigm in (c), and manage to accomplish anomaly detection for various classes with a unified framework in (d).

easing the difficulty of data preparation. Nonetheless, solving such a task is fairly challenging. Recall that the rationale behind unsupervised anomaly detection is to model the distribution of normal data and find a compact decision boundary as in Fig. 1a. When it comes to the multi-class case, we expect the model to capture the distribution of all classes simultaneously such that they can share the same boundary as in Fig. 1b. But if we focus on a particular category, say the green one in Fig. 1b, all the samples from other categories should be considered as anomalies no matter whether they are normal (*i.e.*, blue circles) or anomalous (*i.e.*, blue triangles) themselves. From this perspective, how to accurately model the multi-class distribution becomes vital.

A widely used approach to learning the normal data distribution draws support from image (or feature) reconstruction [2, 5, 26, 39, 51], which assumes that a well-trained model always produces normal samples regardless of the defects within the inputs. In this way, there will be large reconstruction errors for anomalous samples, making them distinguishable from the normal ones. However, we find that popular reconstruction networks suggest unsatisfying performance on the challenging task studied in this work. They typically fall into an "identity shortcut", which appears as returning a direct copy of the input disregarding its content.[1] As a result, even anomalous samples can be well recovered with the learned model and hence become hard to detect. Moreover, under the unified case, where the distribution of normal data is more complex, the "identical shortcut" problem is magnified. Intuitively, to learn a unified model that can reconstruct all kinds of objects, it requires the model to work extremely hard to learn the joint distribution. From this perspective, learning an "identical shortcut" appears as a far easier solution.

To address this issue, we carefully tailor a feature reconstruction framework that prevents the model from learning the shortcut. First, we revisit the formulations of fully-connected layer, convolutional layer, as well as attention layer used in neural networks, and observe that both fully-connected layer and convolutional layer face the risk of learning a trivial solution. This drawback is further amplified under the multi-class setting in that the normal data distribution becomes far more complex. Instead, the attention layer is sheltered from such a risk, benefiting from a learnable query embedding (see Sec. 3.1). Accordingly, we propose a *layer-wise query decoder* to intensify the use of query embedding. Second, we argue that the full attention (*i.e.*, every feature point relates to each other) also contributes to the shortcut issue, because it offers the chance of directly copying the input to the output. To avoid the information leak, we employ a *neighbor masked attention* module, where a feature point relates to neither itself nor its neighbors. Third, inspired by Bengio et al. [3], we propose a *feature jittering* strategy, which requires the model to recover the source message even with noisy inputs. All these designs help the model escape from the "identity shortcut", as shown in Fig. 2b. Extensive experiments on MVTec-AD [4] and CIFAR-10 [23] demonstrate the sufficient superiority of our approach, which we call *UniAD*, over existing alternatives under the unified task setting. For instance, when learning a single model for 15 categories in MVTec-AD, we achieve state-of-the-art performance on the tasks of both anomaly detection and anomaly localization, boosting the AUROC from 88.1% to 96.5% and from 89.5% to 96.8%, respectively.

---

[1]A detailed analysis can be found in Sec. 3.1 and Fig. 2.

## 2 Related work

**Anomaly detection.** *1) Classical approaches* extend classical machine learning methods for one-class classification, such as one-class support vector machine (OC-SVM) [38] and support vector data description (SVDD) [35, 41]. Patch-level embedding [48], geometric transformation [18], and elastic weight consolidation [33] are incorporated for improvement. *2) Pseudo-anomaly* converts anomaly detection to supervised learning, including classification [25, 32, 45], image denoising [52], and hyper-sphere segmentation [27]. However, these methods partly rely on how well proxy anomalies match real anomalies that are not known [13]. *3) Modeling then comparison* assumes that the pre-trained network is capable of extracting discriminative features for anomaly detection [11, 34]. PaDiM [11] and MDND [34] extract pre-trained features to model normal distribution, then utilize a distance metric to measure the anomalies. Nevertheless, these methods need to memorize and model all normal features, thus are computationally expensive. *4) Knowledge distillation* proposes that the student distilled by a teacher on normal samples could only extract normal features [6, 13, 37, 44, 45]. Recent works mainly focus on model ensemble [6], feature pyramid [37, 44], and reverse distillation [13].

**Reconstruction-based anomaly detection.** These methods rely on the hypothesis that reconstruction models trained on normal samples only succeed in normal regions, but fail in anomalous regions [5, 8, 26, 36, 49]. Early attempts include Auto-Encoder (AE) [5, 9], Variational Auto-Encoder (VAE) [22, 26], and Generative Adversarial Net (GAN) [2, 30, 36, 51]. However, these methods face the problem that the model could learn tricks that the anomalies are also restored well. Accordingly, researchers adopt different strategies to tackle this issue, such as adding instructional information (*i.e.*, structural [53] or semantic [39, 46]), memory mechanism [19, 20, 29], iteration mechanism [12], image masking strategy [47], and pseudo-anomaly [9, 32]. Recently, DRAEM [52] first recovers the pseudo-anomaly disturbed normal images for representation, then utilizes a discriminative net to distinguish the anomalies, achieving excellent performance. However, DRAEM [52] ceases to be effective under the unified case. Moreover, there is still an important aspect that has not been well studied, *i.e.*, what architecture is the best reconstruction model? In this paper, we first compare and analyze three popular architectures including MLP, CNN, and transformer. Then, accordingly, we base on the transformer and further design three improvements, which compose our UniAD.

**Transformer in anomaly detection.** Transformer [42] with attention mechanism, first proposed in natural language processing, has been successfully used in computer vision [7, 16]. Some attempts try to utilize transformer for anomaly detection. InTra [31] adopts transformer to recover the image by recovering all masked patches one by one. VT-ADL [28] and AnoVit [50] both apply transformer encoder to reconstruct images. However, these methods directly utilize vanilla transformer, and do not figure out why transformer brings improvement. In contrast, we confirm the efficacy of the query embedding to prevent the shortcut, and accordingly design a layer-wise query decoder. Also, to avoid the information leak of the full attention, we employ a neighbor masked attention module.

## 3 Method

### 3.1 Revisiting feature reconstruction for anomaly detection

In Fig. 2, following the feature reconstruction paradigm [39, 49], we build an MLP, a CNN, and a transformer (with query embedding) to reconstruct the features extracted by a pre-trained backbone. The reconstruction errors represent the anomaly possibility. The architectures of the three networks are given in *Appendix*. The metric is evaluated every 10 epochs. Note that the periodic evaluation is *impractical* since anomalies are not available during training. As shown in Fig. 2a, after a period of training, the performances of the three networks decrease severely with the losses going extremely small. We attribute this to the problem of "identical shortcut", where both normal and anomalous regions can be well recovered, thus failing to spot anomalies. This speculation is verified by the visualization results in Fig. 2b (more results in *Appendix*). However, compared with MLP and CNN, the transformer suffers from a much smaller performance drop, indicating a slighter shortcut problem. This encourages us to analyze as follows.

We denote the features in a normal image as $x^+ \in \mathbb{R}^{K \times C}$, where $K$ is the feature number, $C$ is the channel dimension. The batch dimension is omitted for simplicity. Similarly, the features in an anomalous image are denoted as $x^- \in \mathbb{R}^{K \times C}$. The reconstruction loss is chosen as the MSE loss.

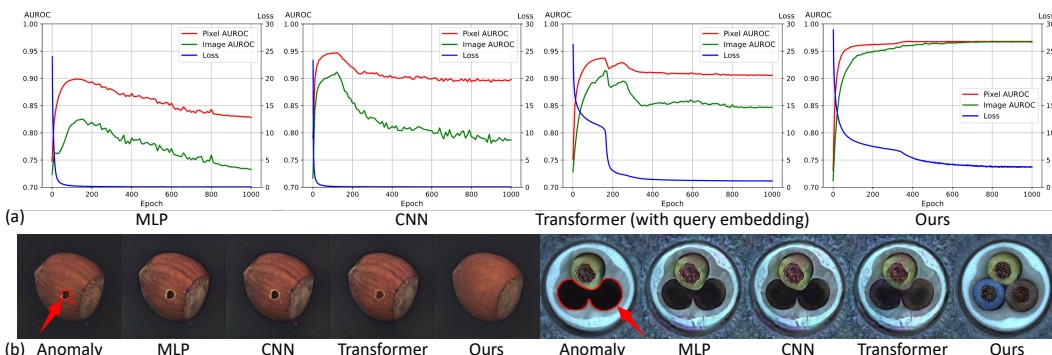

Figure 2: **Comparison among MLP, CNN, transformer, and our UniAD** on MVTec-AD [4]. (a) Training loss (blue) as well as the testing AUROC on anomaly detection (green) and localization (red). During the training of MLP, CNN, and transformer, the reconstruction error keeps going smaller on normal samples, but the performance on anomalies suffers from a severe drop after reaching the peak. This is caused by the model learning an "identical shortcut", which tends to directly copy the input as the output regardless of whether it is normal or anomalous. (b) Visual explanation of the shortcut issue, where the anomalous samples can be well recovered and hence become hard to detect from normal ones. In contrast, UniAD overcomes such a problem and manages to *reconstruct anomalies as normal samples*. It is noteworthy that all models are learned for feature reconstruction and a separate decoder is employed to render images from features. This decoder is *only* used for visualization.

We provide a rough analysis using a simple 1-layer network as the reconstruction net, which is trained with $x^+$ and tested to detect anomalous regions in $x^-$.

**Fully-connected layer in MLP**. Denote the weights and bias in this layer as $w \in \mathbb{R}^{C \times C}, b \in \mathbb{R}^C$, respectively, this layer can be represented as,

$$y = x^+ w + b \in \mathbb{R}^{K \times C}. \tag{1}$$

With the MSE loss pushing $y$ to $x^+$, the model may take shortcut to regress $w \to I$ (identity matrix), $b \to 0$. Ultimately, this model could also reconstruct $x^-$ well, failing in anomaly detection.

**Convolutional layer in CNN**. A convolutional layer with $1 \times 1$ kernel is equivalent to a fully-connected layer. Besides, An $n \times n$ $(n > 1)$ kernel has more parameters and larger capacity, and can complete whatever $1 \times 1$ kernel can. Thus, this layer also has the chance to learn a shortcut.

**Transformer with query embedding**. In such a model, there is an attention layer with a learnable query embedding, $q \in \mathbb{R}^{K \times C}$. When using this layer as the reconstruction model, it is denoted as,

$$y = \mathtt{softmax}(q(x^+)^T/\sqrt{C})x^+ \in \mathbb{R}^{K \times C}. \tag{2}$$

To push $y$ to $x^+$, the attention map, $\mathtt{softmax}(q(x^+)^T/\sqrt{C})$, should approximate $I$ (identity matrix), so $q$ must be highly related to $x^+$. Considering that $q$ in the trained model is relevant to normal samples, the model could not reconstruct $x^-$ well. The ablation study in Sec. 4.6 shows that without the query embedding, the performance of transformer drops dramatically by 18.1% and 13.4% in anomaly detection and localization, respectively. Thus the query embedding is of vital significance to model the normal distribution.

However, transformer still suffers from the shortcut problem, which inspires our three improvements. 1) According to that the query embedding can prevent reconstructing anomalies, we design a Layer-wise Query Decoder (LQD) by adding the query embedding in each decoder layer rather than only the first layer in vanilla transformer. 2) We suspect that the full attention increases the possibility of the shortcut. Since one token could see itself and its neighbor regions, it is easy to reconstruct by simply copying. Thus we mask the neighbor tokens when calculating the attention map, called Neighbor Masked Attention (NMA). 3) We employ a Feature Jittering (FJ) strategy to disturb the input features, leading the model to learn normal distribution from denoising. Benefiting from these designs, our UniAD achieves satisfying performance, as illustrated in Fig. 2.

**Relation between the "identical shortcut" problem and the unified case**. In Fig. 2a, we aim to visualize the "identical shortcut" problem, where the loss becomes smaller yet the performance drops.

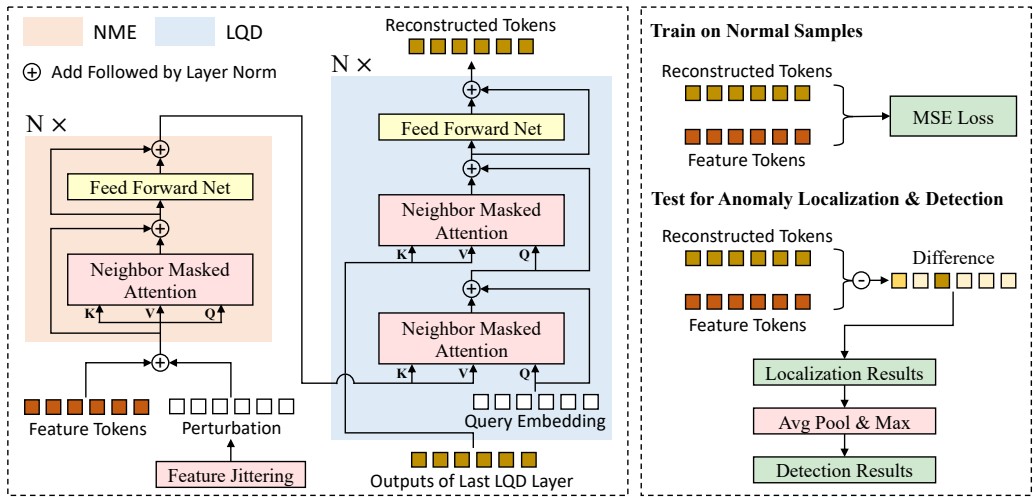

Figure 3: **Framework** of UniAD, consisting of a Neighbor Masked Encoder (NME) and a Layer-wise Query Decoder (LQD). Each layer in LQD employs a *learnable query embedding* to help model the complex training data distribution. The full attention in transformer is replaced by *neighbor masked attention* to avoid the information leak from the input to the output. The *feature jittering* strategy encourages the model to recover the correct message with noisy inputs. All the three improvements assist the model against learning the "identical shortcut" (see Sec. 3.1 and Fig. 2 for details).

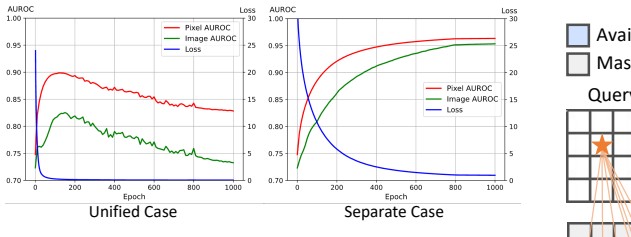

Figure 4: **Comparison between the unified case and the separate case** on the training curves of MLP. In the separate case, the curves are obtained by averaging all categories. Compared with the separate case, the unified case has a smaller reconstruction error but much worse performance, indicating a severer "identical shortcut" problem.

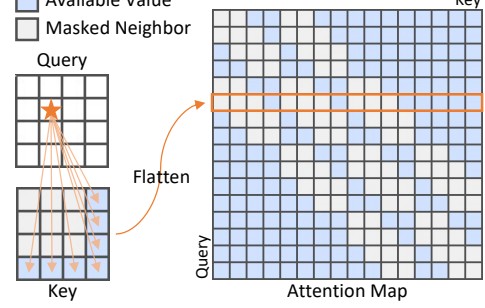

Figure 5: **Illustration of neighbor masked attention**, where a feature point relates to neither itself nor its neighbors.

We conduct the same experiment under the separate case on MLP. As shown in Fig. 4, the accuracy (green for detection and red for localization) keeps growing up along with the loss (blue) getting smaller. This helps reveal the relation between the "identical shortcut" problem and the unified case, which is that *the unified case is more challenging and hence magnifies the "identical shortcut" problem*. Therefore, since our approach is specially designed to solve the "identical shortcut" problem, our method can be effective in the unified case.

## 3.2 Improving feature reconstruction for unified anomaly detection

**Overview**. As shown in Fig. 3, our UniAD is composed of a Neighbor Masked Encoder (NME) and a Layer-wise Query Decoder (LQD). Firstly, the feature tokens extracted by a fixed pre-trained backbone are further integrated by NME to derive the encoder embeddings. Then, in each layer of LQD, a learnable query embedding is successively fused with the encoder embeddings and the outputs of the previous layer (self-fusion for the first layer). The feature fusion is completed by the Neighbor Masked Attention (NMA). The final outputs of LQD are viewed as the reconstructed features. Also, we propose a Feature Jittering (FJ) strategy to add perturbations to the input features, leading the model to learn normal distribution from the denoising task. Finally, the results of anomaly localization and detection are obtained through the reconstruction differences.

**Neighbor masked attention**. We suspect that the full attention in vanilla transformer [42] contributes to the "identical shortcut". In full attention, one token is permitted to see itself, so it will be easy to reconstruct by simply copying. Moreover, considering that the feature tokens are extracted by a CNN backbone, the neighbor tokens must share lots of similarities. Therefore, we propose to mask the neighbor tokens when calculating the attention map, called Neighbor Masked Attention (NMA). Note that the neighbor region is defined in the 2D space, as shown in Fig. 5.

**Neighbor masked encoder**. The encoder follows the standard architecture in vanilla transformer. Each layer consists of an attention module and a Feed-Forward Network (FFN). However, the full attention is replaced by our proposed NMA to prevent the information leak.

**Layer-wise query decoder**. It is analyzed in Sec. 3.1 that the query embedding could help prevent reconstructing anomalies well. However, there is only one query embedding in the vanilla transformer. Therefore, we design a Layer-wise Query Decoder (LQD) to intensify the use of query embedding, as shown in Fig. 3. Specifically, in each layer of LQD, a learnable query embedding is first fused with the encoder embeddings, then integrated with the outputs of the previous layer (self-integration for the first layer). The feature fusion is implemented by NMA. Following the vanilla transformer, a 2-layer FFN is applied to handle these fused tokens, and the residual connection is utilized to facilitate the training. The final outputs of LQD serve as the reconstructed features.

**Feature jittering**. Inspired by Denoising Auto-Encoder (DAE) [3, 43], we add perturbations to feature tokens, guiding the model to learn knowledge of normal samples by the denoising task. Specifically, for a feature token, $\boldsymbol{f}_{tok} \in \mathbb{R}^C$, we sample the disturbance $D$ from a Gaussian distribution,

$$D \sim N(\mu = 0, \sigma^2 = (\alpha \frac{||\boldsymbol{f}_{tok}||_2}{C})^2), \tag{3}$$

where $\alpha$ is the jittering scale to control the noisy degree. Also, the sampled disturbance is added to $\boldsymbol{f}_{tok}$ with a fixed jittering probability, $p$.

## 3.3 Implementation details

**Feature extraction**. We adopt a fixed EfficientNet-b4 [40] pre-trained on ImageNet [14] as the feature extractor. The features from stage-1 to stage-4 are selected. Here the stage means the combination of blocks that have the same size of feature maps. Then these features are resized to the same size, and concatenated along channel dimension to form a feature map, $\boldsymbol{f}_{org} \in \mathbb{R}^{C_{org} \times H \times W}$.

**Feature reconstruction**. The feature map, $\boldsymbol{f}_{org}$, is first tokenized to $H \times W$ feature tokens, followed by a linear projection to reduce $C_{org}$ to a smaller channel, $C$. Then these tokens are processed by NME and LQD. The learnable position embeddings [15, 16] are added in attention modules to inform the spatial information. Afterward, another linear projection is used to recover the channel from $C$ to $C_{org}$. After reshape, the reconstructed feature map, $\boldsymbol{f}_{rec} \in \mathbb{R}^{C_{org} \times H \times W}$, is finally obtained.

**Objective function**. Our model is trained with the MSE loss as,

$$\mathcal{L} = \frac{1}{H \times W}||\boldsymbol{f}_{org} - \boldsymbol{f}_{rec}||_2^2. \tag{4}$$

**Inference for anomaly localization**. The result of anomaly localization is an anomaly score map, which assigns an anomaly score for each pixel. Specifically, the anomaly score map, $s$, is calculated as the L2 norm of the reconstruction differences as,

$$\boldsymbol{s} = ||\boldsymbol{f}_{org} - \boldsymbol{f}_{rec}||_2 \in \mathbb{R}^{H \times W}. \tag{5}$$

Then $s$ is up-sampled to the image size with bi-linear interpolation to obtain the localization results.

**Inference for anomaly detection**. Anomaly detection aims to detect whether an image contains anomalous regions. We transform the anomaly score map, $s$, to the anomaly score of the image by taking the maximum value of the averagely pooled $s$.

## 4 Experiment

### 4.1 Datasets and metrics

**MVTec-AD** [4] is a comprehensive, multi-object, multi-defect industrial anomaly detection dataset with 15 classes. For each anomalous sample in the test set, the ground-truth includes both image

Table 1: **Anomaly detection results with AUROC metric on MVTec-AD** [4]. All methods are evaluated under the unified / separate case. In the unified case, the learned model is applied to detect anomalies for all categories *without* fine-tuning.

| | Category | US [6] | PSVDD [48] | PaDiM [11] | CutPaste [25] | MKD [37] | DRAEM [52] | Ours |
|---|---|---|---|---|---|---|---|---|
| Object | Bottle | 84.0 / 99.0 | 85.5 / 98.6 | 97.9 / 99.9 | 67.9 / 98.2 | 98.7 / 99.4 | 97.5 / 99.2 | **99.7** ± 0.04 / 100 |
| | Cable | 60.0 / 86.2 | 64.4 / 90.3 | 70.9 / 92.7 | 69.2 / 81.2 | 78.2 / 89.2 | 57.8 / 91.8 | **95.2** ± 0.84 / 97.6 |
| | Capsule | 57.6 / 86.1 | 61.3 / 76.7 | 73.4 / 91.3 | 63.0 / 98.2 | 68.3 / 80.5 | 65.3 / 98.5 | **86.9** ± 0.73 / 85.3 |
| | Hazelnut | 95.8 / 93.1 | 83.9 / 92.0 | 85.5 / 92.0 | 80.9 / 98.3 | 97.1 / 98.4 | 93.7 / 100 | **99.8** ± 0.10 / 99.9 |
| | Metal Nut | 62.7 / 82.0 | 80.9 / 94.0 | 88.0 / 98.7 | 60.0 / 99.9 | 64.9 / 73.6 | 72.8 / 98.7 | **99.2** ± 0.09 / 99.0 |
| | Pill | 56.1 / 87.9 | 89.4 / 86.1 | 68.8 / 93.3 | 71.4 / 94.9 | 79.7 / 82.7 | 82.2 / 98.9 | **93.7** ± 0.65 / 88.3 |
| | Screw | 66.9 / 54.9 | 80.9 / 81.3 | 56.9 / 85.8 | 85.2 / 88.7 | 75.6 / 83.3 | **92.0** / 93.9 | 87.5 ± 0.57 / 91.9 |
| | Toothbrush | 57.8 / 95.3 | **99.4** / 100 | 95.3 / 96.1 | 63.9 / 99.4 | 75.3 / 92.2 | 90.6 / 100 | 94.2 ± 0.20 / 95.0 |
| | Transistor | 61.0 / 81.8 | 77.5 / 91.5 | 86.6 / 97.4 | 57.9 / 96.1 | 73.4 / 85.6 | 74.8 / 93.1 | **99.8** ± 0.09 / 100 |
| | Zipper | 78.6 / 91.9 | 77.8 / 97.9 | 79.7 / 90.3 | 93.5 / 99.9 | 87.4 / 93.2 | **98.8** / 100 | 95.8 ± 0.51 / 96.7 |
| Texture | Carpet | 86.6 / 91.6 | 63.3 / 92.9 | 93.8 / 99.8 | 93.6 / 93.9 | 69.8 / 79.3 | 98.0 / 97.0 | **99.8** ± 0.02 / 99.9 |
| | Grid | 69.2 / 81.0 | 66.0 / 94.6 | 73.9 / 96.7 | 93.2 / 100 | 83.8 / 78.0 | **99.3** / 99.9 | 98.2 ± 0.26 / 98.5 |
| | Leather | 97.2 / 88.2 | 60.8 / 90.9 | 99.9 / 100 | 93.4 / 100 | 93.6 / 95.1 | 98.7 / 100 | **100** ± 0.00 / 100 |
| | Tile | 93.7 / 99.1 | 88.3 / 97.8 | 93.3 / 98.1 | 88.6 / 94.6 | 89.5 / 91.6 | **99.8** / 99.6 | 99.3 ± 0.14 / 99.0 |
| | Wood | 90.6 / 97.7 | 72.1 / 96.5 | 98.4 / 99.2 | 80.4 / 99.1 | 93.4 / 94.3 | **99.8** / 99.1 | 98.6 ± 0.08 / 97.9 |
| | Mean | 74.5 / 87.7 | 76.8 / 92.1 | 84.2 / 95.5 | 77.5 / 96.1 | 81.9 / 87.8 | 88.1 / 98.0 | **96.5** ± 0.08 / 96.6 |

label and anomaly segmentation. In the existing literature, only the separate case is researched. In this paper, we introduce the unified case, where only one model is used to handle all categories.

**CIFAR-10** [23] is a classical image classification dataset with 10 categories. Existing methods [6, 24, 37] evaluate CIFAR-10 mainly in the *one-versus-many* setting, where one class is viewed as normal samples, and others serve as anomalies. Semantic AD [1, 10] proposes a *many-versus-one* setting, treating one class as anomalous and the remaining classes as normal. Different from both, we propose a unified case (*many-versus-many* setting), which is detailed in Sec. 4.4.

**Metrics**. Following prior works [4, 6, 52], the Area Under the Receiver Operating Curve (AUROC) is used as the evaluation metric for anomaly detection.

## 4.2 Anomaly detection on MVTec-AD

**Setup**. Anomaly detection aims to detect whether an image contains anomalous regions. The anomaly detection performance is evaluated on MVTec-AD [4]. The image size is selected as $224 \times 224$, and the size for resizing feature maps is set as $14 \times 14$. The feature maps from stage-1 to stage-4 of EfficientNet-b4 [40] are resized and concatenated together to form a 272-channel feature map. The reduced channel dimension is set as 256. AdamW optimizer [21] with weight decay $1 \times 10^{-4}$ is used. Our model is trained for 1000 epochs on 8 GPUs (NVIDIA Tesla V100 16GB) with batch size 64. The learning rate is $1 \times 10^{-4}$ initially, and dropped by 0.1 after 800 epochs. The layer numbers of the encoder and decoder are both 4. The neighbor size, jittering scale, and jittering probability are set as $7 \times 7$, 20, and 1, respectively. The evaluation is run with 5 random seeds. In both the separate case and the unified case, the reconstruction models are trained from the scratch.

**Baselines**. Our approach is compared with baselines including: US [6], PSVDD [48], PaDiM [11], CutPaste [25], MKD [37], and DRAEM [52]. Under the separate case, the baselines' metric is reported in their papers except the metric of US borrowed from [52]. Under the unified case, US, PSVDD, PaDiM, CutPaste, MKD, and DRAEM are run with the publicly available implementations.

**Quantitative results of anomaly detection on MVTec-AD** [4] are shown in Tab. 1. Though all baselines achieve excellent performances under the separate case, their performances drop dramatically under the unified case. The previous SOTA, DRAEM, a reconstruction-based method trained by pseudo-anomaly, suffers from a drop of near 10%. For another strong baseline, CutPaste, a pseudo-anomaly approach, the drop is as large as 18.6%. However, our UniAD has almost no performance drop from the separate case (96.6%) to the unified case (96.5%). Moreover, we beat the best competitor, DRAEM, by a dramatically large margin (8.4%), demonstrating our superiority.

Table 2: **Anomaly localization results with AUROC metric on MVTec-AD** [4]. All methods are evaluated under the unified / separate case. In the unified case, the learned model is applied to detect anomalies for all categories *without* fine-tuning.

| | Category | US [6] | PSVDD [48] | PaDiM [11] | FCDD [27] | MKD [37] | DRAEM [52] | Ours |
|---|---|---|---|---|---|---|---|---|
| Object | Bottle | 67.9 / 97.8 | 86.7 / 98.1 | 96.1 / 98.2 | 56.0 / 97 | 91.8 / 96.3 | 87.6 / 99.1 | **98.1** ± 0.04 / 98.1 |
| | Cable | 78.3 / 91.9 | 62.2 / 96.8 | 81.0 / 96.7 | 64.1 / 90 | 89.3 / 82.4 | 71.3 / 94.7 | **97.3** ± 0.10 / 96.8 |
| | Capsule | 85.5 / 96.8 | 83.1 / 95.8 | 96.9 / 98.6 | 67.6 / 93 | 88.3 / 95.9 | 50.5 / 94.3 | **98.5** ± 0.01 / 97.9 |
| | Hazelnut | 93.7 / 98.2 | 97.4 / 97.5 | 96.3 / 98.1 | 79.3 / 95 | 91.2 / 94.6 | 96.9 / 99.7 | **98.1** ± 0.10 / 98.8 |
| | Metal Nut | 76.6 / 97.2 | **96.0** / 98.0 | 84.8 / 97.3 | 57.5 / 94 | 64.2 / 86.4 | 62.2 / 99.5 | 94.8 ± 0.09 / 95.7 |
| | Pill | 80.3 / 96.5 | **96.5** / 95.1 | 87.7 / 95.7 | 65.9 / 81 | 69.7 / 89.6 | 94.4 / 97.6 | 95.0 ± 0.16 / 95.1 |
| | Screw | 90.8 / 97.4 | 74.3 / 95.7 | 94.1 / 98.4 | 67.2 / 86 | 92.1 / 96.0 | 95.5 / 97.6 | **98.3** ± 0.08 / 97.4 |
| | Toothbrush | 86.9 / 97.9 | 98.0 / 98.1 | 95.6 / 98.8 | 60.8 / 94 | 88.9 / 96.1 | 97.7 / 98.1 | **98.4** ± 0.03 / 97.8 |
| | Transistor | 68.3 / 73.7 | 78.5 / 97.0 | 92.3 / 97.6 | 54.2 / 88 | 71.7 / 76.5 | 64.5 / 90.9 | **97.9** ± 0.19 / 98.7 |
| | Zipper | 84.2 / 95.6 | 95.1 / 95.1 | 94.8 / 98.4 | 63.0 / 92 | 86.1 / 93.9 | **98.3** / 98.8 | 96.8 ± 0.24 / 96.0 |
| Texture | Carpet | 88.7 / 93.5 | 78.6 / 92.6 | 97.6 / 99.0 | 68.6 / 96 | 95.5 / 95.6 | **98.6** / 95.5 | 98.5 ± 0.01 / 98.0 |
| | Grid | 64.5 / 89.9 | 70.8 / 96.2 | 71.0 / 97.1 | 65.8 / 91 | 82.3 / 91.8 | **98.7** / 99.7 | 96.5 ± 0.04 / 94.6 |
| | Leather | 95.4 / 97.8 | 93.5 / 97.4 | 84.8 / 99.0 | 66.3 / 98 | 96.7 / 98.1 | 97.3 / 98.6 | **98.8** ± 0.03 / 98.3 |
| | Tile | 82.7 / 92.5 | 92.1 / 91.4 | 80.5 / 94.1 | 59.3 / 91 | 85.3 / 82.8 | **98.0** / 99.2 | 91.8 ± 0.10 / 91.8 |
| | Wood | 83.3 / 92.1 | 80.7 / 90.8 | 89.1 / 94.1 | 53.3 / 88 | 80.5 / 84.8 | **96.0** / 96.4 | 93.2 ± 0.08 / 93.4 |
| | Mean | 81.8 / 93.9 | 85.6 / 95.7 | 89.5 / 97.4 | 63.3 / 92 | 84.9 / 90.7 | 87.2 / 97.3 | **96.8** ± 0.02 / 96.6 |

Figure 6: **Qualitative results** for anomaly localization on MVTec-AD [4]. From left to right: normal sample as the reference, anomaly, our reconstruction, ground-truth, and our predicted anomaly map. The approach to visualizing reconstruction is the same as the one used in Fig. 2.

## 4.3 Anomaly localization on MVTec-AD

**Setup and baselines**. Anomaly localization aims to localize anomalous regions in an anomalous image. MVTec-AD [4] is chosen as the benchmark dataset. The setup is the same as that in Sec. 4.2. Besides the competitors in Sec. 4.2, FCDD [27] is included, whose metric under the separate case is reported in its paper. Under the unified case, we run FCDD with the implementation: FCDD.

**Quantitative results of anomaly localization on MVTec-AD** [4] are reported in Tab. 2. Similar to Sec. 4.2, switching from the separate case to the unified case, the performance of all competitors drops significantly. For example, the performance of US, an important distillation-based baseline, decreases by 12.1%. FCDD, a pseudo-anomaly approach, suffers from a dramatic drop of 28.7%, reflecting the pseudo-anomaly is not suitable for the unified case. However, our UniAD even gains a slight improvement from the separate case (96.6%) to the unified case (96.8%), proving the suitability of our UniAD for the unified case. Moreover, we significantly surpass the strongest baseline, PaDiM, by 7.3%. This significant improvement reflects the effectiveness of our model.

**Qualitative results for anomaly localization on MVTec-AD** [4] are illustrated in Fig. 6. For both global (Fig. 6a) and local (Fig. 6b) structural anomalies, both scattered texture perturbations (Fig. 6c) and multiple texture scratches (Fig. 6d), our method could successfully reconstruct anomalies to their corresponding normal samples, then accurately localize anomalous regions through reconstruction differences. More qualitative results are given in *Appendix*.

## 4.4 Anomaly detection on CIFAR-10

**Setup**. To further verify the effectiveness of our UniAD, we extend CIFAR-10 [23] to the unified case, which consists of four combinations. For each combination, five categories together serve as normal samples, while other categories are viewed as anomalies. The class indices of the four

Table 3: **Anomaly detection results with AUROC metric on CIFAR-10** [23] under the unified case. Here, {01234} means samples from class 0, 1, 2, 3, 4 are borrowed as the normal ones.

| Normal Indices | US [6] | FCDD [27] | FCDD+OE [27] | PANDA [33] | MKD [37] | Ours |
|---|---|---|---|---|---|---|
| {01234} | 51.3 | 55.0 | 71.8 | 66.6 | 64.2 | **84.4** $\pm$ 0.02 |
| {56789} | 51.3 | 50.3 | 73.7 | 73.2 | 69.3 | **80.9** $\pm$ 0.02 |
| {02468} | 63.9 | 59.2 | 85.3 | 77.1 | 76.4 | **93.0** $\pm$ 0.03 |
| {13579} | 56.8 | 58.5 | 85.0 | 72.9 | 78.7 | **90.6** $\pm$ 0.09 |
| Mean | 55.9 | 55.8 | 78.9 | 72.4 | 72.1 | **87.2** $\pm$ 0.03 |

Table 4: **Performance comparison and architecture comparison** between UniAD and transformer-based competitors on MVTec-AD [4]. All methods are evaluated under the unified / separate case.

| Method | Det. | Loc. | 1 query | Layer-wise query |
|---|---|---|---|---|
| InTra [31] | 65.3 / 95.0 | 70.6 / 96.6 | ✗ | ✗ |
| VT-ADL [28] | 55.4 / 78.7 | 64.4 / 82.0 | ✗ | ✗ |
| AnoVit [50] | 69.6 / 78 | 68.4 / 83 | ✗ | ✗ |
| Ours (baseline) | 87.6 / 94.7 | 92.8 / 95.8 | ✓ | ✗ |
| Ours | **96.5** / 96.6 | **96.8** / 96.6 | ✗ | ✓ |

combinations are {01234}, {56789}, {02468}, {13579}. Here, {01234} means the normal samples include images from class 0, 1, 2, 3, 4, and similar for others. Note that the class index is obtained by sorting the class names of 10 classes. The setup of the model is detailed in *Appendix*.

**Baselines**. US [6], FCDD [27], FCDD+OE [27], PANDA [33], and MKD [37] serve as competitors. US, FCDD, FCDD+OE, PANDA, and MKD are run with the publicly available implementations.

**Quantitative results of anomaly detection on CIFAR-10** [23] are shown in Tab. 3. When five classes together serve as normal samples, two recent baselines, US and FCDD, almost lose their ability to detect anomalies. When utilizing 10000 images sampled from CIFAR-100 [23] as auxiliary Outlier Exposure (OE), FCDD+OE improves the performance by a large margin. We still stably outperform FCDD+OE by 8.3% without the help of OE, indicating the efficacy of our UniAD.

## 4.5 Comparison with transformer-based competitors

As described in Sec. 2, some attempts [31, 28, 50] also try to utilize transformer for anomaly detection. Here we compare our UniAD with existing transformer-based competitors on MVTec-AD [4]. Recall that, we choose transformer as the reconstruction model considering its great potential in preventing the model from learning the "identical shortcut" (refer to Sec. 3.1). Concretely, we find that the *learnable query embedding* is essential for avoiding such a shortcut but is seldom explored in existing transformer-based approaches. As shown in Tab. 4, after introducing even only one query embedding, our baseline already outperforms existing alternatives by a sufficiently large margin in the unified setting. Our proposed three components further improve our *strong baseline*. Recall that all three components are proposed to avoid the model from directly outputting the inputs.

## 4.6 Ablation studies

To verify the effectiveness of the proposed modules and the selection of hyperparameters, we implement extensive ablation studies on MVTec-AD [4] under the unified case.

**Layer-wise query**. Tab. 5a verifies our assertion that the query embedding is of vital significance. 1) Without query embedding, meaning the encoder embeddings are directly input to the decoder, the performance is the worst. 2) Adding only one query embedding to the first decoder layer (*i.e.*, vanilla transformer [42]) promotes the performance dramatically by 18.1% and 13.4% in anomaly detection and localization, respectively. 3) With layer-wise query embedding in each decoder layer, image-level and pixel-level AUROC is further improved by 7.4% and 3.7%, respectively.

**Layer number**. We conduct experiments to investigate the influence of layer number, as shown in Tab. 5b. 1) No matter with which combination, our model outperforms vanilla transformer by a large margin, reflecting the effectiveness of our design. 2) The best performance is achieved with a

Table 5: **Ablation studies with AUROC metric on MVTec-AD** [4]. Default settings are in blue.

(a) Layer-wise query, NMA, & FJ

| w/o q. | 1 q. | Layer-wise q. | NMA | FJ | Det. | Loc. |
|---|---|---|---|---|---|---|
| ✓ | - | - | - | - | 69.5 | 79.4 |
| - | ✓ | - | - | - | 87.6 | 92.8 |
| - | - | ✓ | - | - | 95.0 | 96.5 |
| - | ✓ | - | ✓ | - | 96.1 | 96.3 |
| - | ✓ | - | - | ✓ | 95.0 | 95.8 |
| - | - | ✓ | ✓ | ✓ | **96.5** | **96.8** |

(b) Layer Number of Encoder & Decoder

| #Enc, #Dec | Vanilla [42] Det. | Loc. | Ours Det. | Loc. |
|---|---|---|---|---|
| 4, 0 | 69.8 | 79.2 | 94.9 | 96.0 |
| 0, 4 | 80.5 | 88.3 | 96.1 | 96.3 |
| 2, 2 | 84.7 | 90.6 | 95.1 | 96.0 |
| 4, 4 | 87.6 | 92.8 | **96.5** | **96.8** |
| 6, 6 | 86.1 | 91.9 | **96.5** | 96.7 |

(c) Neighbor Size in NMA

| Size | Det. | Loc. |
|---|---|---|
| 1×1 | 94.6 | 96.3 |
| 5×5 | 96.4 | **96.8** |
| 7×7 | **96.5** | **96.8** |
| 9×9 | 96.3 | 96.7 |

(d) Where to Add NMA

| Place | Det. | Loc. |
|---|---|---|
| Enc | 95.8 | 96.3 |
| Enc+Dec1 | 96.4 | **96.8** |
| Enc+Dec2 | **96.5** | 96.7 |
| All | **96.5** | **96.8** |

(e) Jitter Scale $\alpha$ in FJ

| $\alpha$ | Det. | Loc. |
|---|---|---|
| 5 | 96.1 | 96.7 |
| 10 | 96.4 | 96.7 |
| 20 | **96.5** | **96.8** |
| 30 | 95.7 | 96.6 |

(f) Jitter Prob. $p$ in FJ

| $p$ | Det. | Loc. |
|---|---|---|
| 0.25 | 95.6 | 96.5 |
| 0.50 | 95.8 | 96.7 |
| 0.75 | 96.3 | 96.7 |
| 1 | **96.5** | **96.8** |

moderate layer number: 4Enc+4Dec. A larger layer number like 6Enc+6Dec does not bring further promotion, which may be because more layers are harder to train.

**Neighbor masked attention**. 1) The effectiveness of NMA is proven in Tab. 5a. Under the case of one query embedding, adding NMA brings promotion by 8.5% for detection and 3.5% for localization. 2) The neighbor size of NMA is selected in Tab. 5c. 1×1 neighbor size is the worst, because 1×1 is too small to prevent the information leak, thus the recovery could be completed by copying neighbor regions. A larger neighbor size ($\geq$ 5×5) is obviously much better, and the best one is selected as 7×7. 3) We also study the place to add NMA in Tab. 5d. Only adding NMA in the encoder (Enc) is not enough. The performance could be stably improved when further adding NMA in the first or second attention in the decoder (Enc+Dec1, Enc+Dec2) or both (All). This reflects that the full attention of the decoder also contributes to the information leak.

**Feature jittering**. 1) Tab. 5a confirms the efficacy of FJ. With one query embedding as the baseline, introducing FJ could bring an increase of 7.4% for detection and 3.0% for localization, respectively. 2) According to Tab. 5e, the jittering scale, $\alpha$, is chosen as 20. A larger $\alpha$ (*i.e.*, 30) disturbs the feature too much, degrading the results. 3) In Tab. 5f, the jittering probability, $p$, is studied. In essence, the task would be a denoising task with feature jittering, and be a reconstruction task without feature jittering. The results show that the full denoising task (*i.e.*, $p = 1$) is the best.

## 5 Conclusion

In this work, we propose UniAD that unifies anomaly detection regarding multiple classes. For such a challenging task, we assist the model against learning an "identical shortcut" with three improvements. First, we confirm the effectiveness of the learnable query embedding and carefully tailor a layer-wise query decoder to help model the complex distribution of multi-class data. Second, we come up with a neighbor masked attention module to avoid the information leak from the input to the output. Third, we propose feature jittering that helps the model less sensitive to the input perturbations. Under the unified task setting, our method achieves state-of-the-art performance on MVTec-AD and CIFAR-10 datasets, significantly outperforming existing alternatives.

**Discussion**. In this work, different kinds of objects are handled without being distinguished. We have not used the category labels that may help the model better fit multi-class data. How to incorporate the unified model with category labels should be further studied. In practical uses, normal samples are not as consistent as those in MVTec-AD, often manifest themselves in some diversity. Our UniAD could handle all 15 categories in MVTec-AD, hence would be more suitable for real scenes. However, anomaly detection may be used for video surveillance, which may infringe personal privacy.

## Acknowledgments and Disclosure of Funding

**Acknowledgement.** This work is sponsored by the National Key Research and Development Program of China (2021YFB1716000) and National Natural Science Foundation of China (62176152).

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
