# A Unified Model for Multi-class Anomaly Detection
## – *Supplementary Material* –

**Zhiyuan You**[1*]  **Lei Cui**[2*]  **Yujun Shen**[3]  **Kai Yang**[4]  **Xin Lu**[4]  **Yu Zheng**[1]  **Xinyi Le**[1†]

[1]Shanghai Jiao Tong University    [2]Tsinghua University    [3]CUHK    [4]SenseTime

zhiyuanyou@foxmail.com, cuil19@mails.tsinghua.edu.cn, shenyujun0302@gmail.com
{yangkai, luxin}@sensetime.com, yuzheng@sjtu.edu.cn, lexinyi@sjtu.edu.cn

## A  Overview

We organize the supplementary material as follows. Sec. B discusses the setting of our proposed unified case. Sec. C describes the detailed architecture of the three baseline reconstruction networks (MLP, CNN, and transformer), our UniAD, and other assistant modules, followed by the training configurations. Sec. D conducts comprehensive ablation studies on the components of our approach. Sec. E presents more visualization results of the reconstructed features and qualitative results for anomaly localization on MVTec-AD [2] under the unified case.

## B  Discussions about the unified setting

Semantic AD [1, 4] also involves multiple classes, which mainly focus on anomaly detection extended from classification datasets (*e.g.*, CIFAR-10 [9]). Semantic AD treats one class as anomalous and the remaining classes as normal. Our task setting clearly differs from semantic AD.

First, we focus more on the industrial anomaly detection dataset, MVTec-AD [2], which is of more practical usage. Unlike CIFAR-10, *each category has normal and abnormal samples* in MVTec-AD. We would like to model the *joint distribution of normal samples across all categories*. It requires the model to learn "what normal samples from each category look like" instead of "what categories are normal". The latter is the main focus of semantic AD.

Second, we also differ from semantic AD regarding the CIFAR-10 task setting. Semantic AD studies the *many-versus-one* setting, which treats 9 classes as normal and the remaining class as anomalous. In contrast, we study the *many-versus-many* setting, which treats 5 classes as normal and the other 5 classes as anomalous. We use such a setting to simulate the real scenario, where *both normal and anomalous samples contain multiple classes*.

## C  Network architecture and training configurations

### C.1  Reconstruction baselines

We present the architectures of the three reconstruction baselines as follows. These baselines share the same assistant modules and training configurations as our UniAD, which are provided in Sec. C.3 and Sec. C.4.

**CNN** is designed based on the ResNet-34 [7] by revising the followings. 1) We remove the operations before stage-1 (a 7×7 convolutional layer followed by batch normalization, ReLU activation, and

---

* Contribute Equally.    † Corresponding Author.

36th Conference on Neural Information Processing Systems (NeurIPS 2022).

max-pooling). 2) All strides from stage-1 to stage-4 are 1, meaning the size of the feature map is the same. 3) The channel dimensions from stage-1 to stage-4 are respectively $C$, $C/2$, $C/2$, and $C$, where $C$ is the reduced channel dimension that is 256.

**Transformer** follows the architecture of the vanilla transformer [14] with a 4-layer encoder and a 4-layer decoder. 1) Each encoder layer is composed of a self-attention layer and a feed-forward network. 2) Each decoder layer consists of a self-attention layer, a cross-attention layer, and a feed-forward network. For the first decoder layer, the inputs of the self-attention layer are the learnable query embeddings. While, for other decoder layers, the inputs of the self-attention layer are the outputs of the previous decoder layer. The outputs of the self-attention layer serve as the *query* of the cross-attention layer, and the encoder embeddings are used as the *key* and *value* of the cross-attention layer. 3) Like the vanilla transformer, the residual connection is utilized in the attention module and the feed-forward network. 4) The learnable position embeddings [6] are added in all attention modules to inform the spatial information. 5) The feed-forward network is the same as Tab. S1.

**MLP** is revised from the transformer by substituting all attention layers. 1) The self-attention is replaced by a linear projection, followed by layer normalization and ReLU activation. 2) The cross-attention between two sets of inputs is changed to a concatenation along the channel dimension and a linear projection, followed by layer normalization and ReLU activation. 3) The position embeddings are removed because MLP could keep spatial information.

## C.2 UniAD

The whole architecture of our UniAD including the neighbor masked encoder and the layer-wise query decoder has been described in the main paper. Here, the detailed architecture of the feed-forward network is provided in Tab. S1.

Table S1: **Architecture of feed-forward network.**

| layer | dim_in | dim_out | activation |
|---|---|---|---|
| linear_1 | 256 | 1024 | ReLU |
| linear_2 | 1024 | 256 | - |

## C.3 Assistant modules

**Feature extraction**. As stated in the main paper, the selected features are resized to the same size, and concatenated along the channel dimension to form a feature map, $\boldsymbol{f}_{org} \in \mathbb{R}^{C_{org} \times H \times W}$. Afterward, this feature map is tokenized to $H \times W$ feature tokens with $C_{org}$ channels (no tokenization for CNN).

**Channel reduction**. A linear projection (or $1 \times 1$ convolution for CNN) is first applied to reduce $C_{org}$ to a smaller channel, $C$. Then these features are processed by the reconstruction model, followed by another linear projection to recover the channel from $C$ to $C_{org}$. Through reshape (CNN does not need reshape), the reconstructed feature map, $\boldsymbol{f}_{rec} \in \mathbb{R}^{C_{org} \times H \times W}$, is finally obtained.

**Visualization of reconstructed features**. We employ a feature reconstruction paradigm, which is harder to visualize compared with image reconstruction. To intuitively explain the problem of "identical shortcut", we must render these reconstructed features into images. Therefore, we pre-train a decoder on both normal and anomalous samples to recover the backbone-extracted features to images, then use this decoder to render reconstructed features. Note that this decoder is *only* used for visualization. The decoder follows a reversed architecture of ResNet-34 [7]. The down-sampling in the original network is replaced by up-sampling implemented by the transposed convolution.

## C.4 Training configurations on CIFAR-10

**CIFAR-10** [9]. The image size and feature size are set as $224 \times 224$ and $14 \times 14$, respectively. Considering that the anomalies in CIFAR-10 are semantically different objects (not structural damages or texture perturbations in MVTec-AD [2]), the features in deep layers containing more semantic information must be helpful. Therefore, the feature maps from stage-1 to stage-5 are selected. These features are resized and concatenated together to form a 720-channel feature map. The reduced channel dimension is set as 256. Our model is trained for 1000 epochs on 8 GPUs (NVIDIA Tesla V100 16GB) with batch size 128 by AdamW optimizer [8] (with weight decay $1 \times 10^{-4}$). The learning rate is $1 \times 10^{-4}$ initially, and dropped by 0.1 after 800 epochs. The layer numbers of the encoder and decoder are both 4. The neighbor size, jittering scale, and jittering probability are chosen as $7 \times 7$, 20, and 1, respectively. The evaluation is run with 5 random seeds.

Table S2: **Ablation studies regarding layer-wise query embedding, neighbor masked attention (NMA), and feature jittering (FJ)**. Default settings are in blue.

| w/o query | 1 query | layer-wise query | NMA | FJ | Det. | Loc. |
|:---:|:---:|:---:|:---:|:---:|:---:|:---:|
| ✓ | - | - | - | - | 69.5 | 79.4 |
| - | ✓ | - | - | - | 87.6 | 92.8 |
| - | ✓ | - | ✓ | - | 96.1 | 96.3 |
| - | ✓ | - | - | ✓ | 95.0 | 95.8 |
| - | ✓ | - | ✓ | ✓ | 96.2 | 96.6 |
| - | - | ✓ | - | - | 95.0 | 96.5 |
| - | - | ✓ | ✓ | - | 95.8 | 96.5 |
| - | - | ✓ | - | ✓ | 94.9 | 96.2 |
| - | - | ✓ | ✓ | ✓ | **96.5** | **96.8** |

# D  Ablation studies

In this part, we make comprehensive analyses on different components of our UniAD. All experiments are implemented on MVTec-AD [2] and evaluated with AUROC under the unified case.

## D.1  Full ablation studies of our three designs

Because of the page limitation, we do not include the full ablation experiments in the main paper. Here we provide the full ablation studies regarding layer-wise query embedding, neighbor masked attention (NMA), and feature jittering (FJ) in Tab. S2.

Based on the vanilla transformer with one query embedding (1 query), adding NMA or FJ both could obviously improve the results. NMA together with FJ provides a quite strong performance (96.2% for detection and 96.6% for localization). Therefore, the effectiveness and the combined effect of NMA and FJ are verified.

When it comes to the layer-wise query embedding, adding NMA brings a slight promotion, while adding FJ makes the performance slightly worse. Adding both NMA and FJ could achieve the best results (96.5% for detection and 96.8% for localization). These reflect that, under the layer-wise query embedding, FJ must cooperate with NMA to function, and combining all three components could bring the best performance.

## D.2  Layer-wise query decoder

For each decoder layer (except the first one), there are three inputs, the learnable query embedding, the encoder embedding, and the outputs of the previous layer. These three sources of information should be fused by two attention modules. The learnable query embedding must serve as the *query* of an attention module, while others are not sure. Therefore, there are six combinations in total, as illustrated in Fig. S1. The performances of the six architectures are given in Tab. S3.

First, we focus on the situation where the Neighbor Masked Attention (NMA) and Feature Jittering (FJ) are not adopted. As shown in Tab. S3, (a) achieves the best results, and (a), (b), and (f) all outperform the vanilla transformer [14]. These three designs share some common characteristics as followings. 1) The outputs of the previous layer should be input as the *key* and *value* of the attention module. The reason might be that being *key* and *value* helps aggregate the information layer by layer. 2) If the query embedding is input to the first attention, the results of the first attention should serve as the *query* of the second attention. We assert that the results of the first attention are semantic-instructed query embedding, which functions the same as the query embedding and should be the *query* of the second attention.

Then, we add NMA and FJ to the three designs including (a), (b), and (f) that have been proven effective by above experiments. Adding NMA and FJ promotes the performance stably. (a) keeps the best performance, and is our final choice.

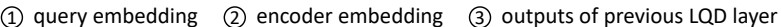

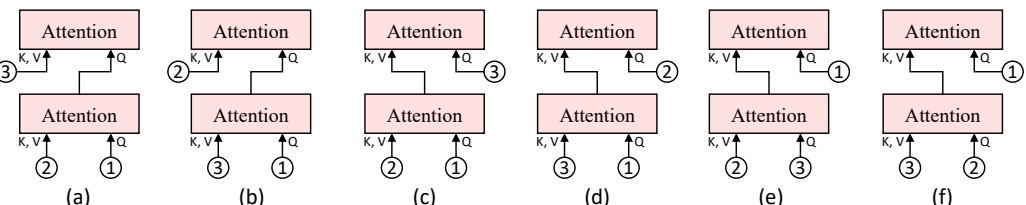

Figure S1: **Various design choices of the Layer-wise Query Decoder (LQD)**, where two attention modules are employed in each layer. The residual connection, layer normalization, and feed-forward network are omitted for simplicity. The performance comparison can be found in Tab. S3.

Table S3: **Ablation study on the design choice of our Layer-wise Query Decoder (LQD)**. Concept of each design can be found in Fig. S1. Performances on anomaly *detection / localization* are reported.

| Vanilla | NMA | FJ | (a) | (b) | (c) | (d) | (e) | (f) |
|---|---|---|---|---|---|---|---|---|
| 87.6 / 92.8 | - | - | **95.0 / 96.5** | 94.4 / 96.3 | 78.4 / 87.0 | 77.9 / 87.0 | 78.2 / 89.9 | 89.8 / 94.0 |
| | ✓ | ✓ | **96.5 / 96.8** | 96.3 / 96.7 | - | - | - | 96.0 / 96.6 |

## D.3 Object function

This section studies the loss function used for feature regression. Here we denote the original features as $\boldsymbol{f}_{org} \in \mathbb{R}^{C_{org} \times H \times W}$, the reconstructed features as $\boldsymbol{f}_{rec} \in \mathbb{R}^{C_{org} \times H \times W}$.

**MSE loss** is one of the most popular regression or reconstruction loss functions. It is represented as,

$$\mathcal{L} = \frac{1}{H \times W} ||\boldsymbol{f}_{org} - \boldsymbol{f}_{rec}||_2^2. \tag{S1}$$

**Normalized MSE loss** adds a normalization to both feature maps, such that each feature vector in the two feature maps is a unit vector. The normalized MSE loss is written as,

$$\mathcal{L} = \frac{1}{H \times W} \left|\left| \frac{\boldsymbol{f}_{org}}{||\boldsymbol{f}_{org}||_2} - \frac{\boldsymbol{f}_{rec}}{||\boldsymbol{f}_{rec}||_2} \right|\right|_2^2. \tag{S2}$$

Also, when we adopt this loss function, the anomaly localization results should be changed to the L2 norm of the differences between two normalized features.

**Cosine distance loss** is used to minimize the cosine distance between the original features and the reconstructed features. It is denoted as,

$$\mathcal{L} = \frac{1}{H \times W} \sum \cos(\boldsymbol{f}_{org}, \boldsymbol{f}_{rec}). \tag{S3}$$

Similarly, to get compatible with the loss function, the anomaly localization results are also obtained through cosine distance.

The performances of the three loss functions are provided in Tab. S4. The three loss functions achieve similar results, proving *the universality of our UniAD* with different loss functions. We finally choose MSE loss because it is the most commonly adopted regression or reconstruction loss function.

Table S4: **Ablation study on the loss function.**

| MSE | | Norm MSE | | Cosine | |
|---|---|---|---|---|---|
| Det. | Loc. | Det. | Loc. | Det. | Loc. |
| 96.5 | 96.8 | 96.4 | **96.9** | **96.6** | 96.8 |

## D.4 Backbone

**Trainable or frozen**. As shown in Tab. S5, if we train the backbone like other modules, the performance drops dramatically on both anomaly detection (-30.6%) and localization (-31.3%). We speculate that training backbone would lead the backbone to extract some indiscriminative features that are easy to reconstruct, which however does not help detect anomalies.

Table S5: **Ablation study on whether to freeze the backbone.**

| Training | | Freezing | |
|---|---|---|---|
| Det. | Loc. | Det. | Loc. |
| 65.9 | 65.5 | **96.5** | **96.8** |

Table S6: **Ablation study on the backbone architecture.** Performances on anomaly *detection / localization* are reported.

| Res-18 11.4M | Res-34 21.5M | Res-50 25.6M | Res-101 44.5M | Eff-b0 5.3M | Eff-b2 9.2M | Eff-b4 19M | Eff-b6 43M |
|---|---|---|---|---|---|---|---|
| 92.4 / 95.8 | 93.0 / 96.2 | 92.4 / 96.0 | 92.2 / 95.9 | 96.1 / 96.4 | 96.2 / **97.0** | **96.5** / 96.8 | 96.0 / 96.7 |

Table S7: **Complexity comparison** between our UniAD and other baselines.

|  | US [3] | PSVDD [15] | PaDiM [5] | CutPaste [10] | FCDD [11] | MKD [12] | DRAEM [16] | Ours |
|---|---|---|---|---|---|---|---|---|
| FLOPs(G) | 60.32 | 149.74 | 23.25 | 3.65 | 13.16 | 32.11 | 245.15 | 6.46 |
| Learnable Params(M) | 9.55 | 0.41 | 950.36 | 13.61 | 4.51 | 0.34 | 69.05 | 7.48 |

**Backbone architecture**. We evaluate two types of backbones, *i.e.*, ResNet [7] and EfficientNet [13]. Both are pre-trained on ImageNet. From Tab. S6, we have the following observations: 1) EfficientNet performs obviously better than ResNet, especially in anomaly detection. 2) The backbone with moderate parameter size is more suitable for anomaly detection, like ResNet-34 in ResNet, and EfficientNet-b2 or EfficientNet-b4 in EfficientNet. The reason might be that too shallow networks could not extract discriminative features, while too deep networks focus more on semantic features, rather than the structural damages or texture perturbations in anomaly detection. We select EfficientNet-b4 as the backbone by default.

# E    More results

## E.1    Complexity comparison

With the image size fixed as $224 \times 224$, we compare the inference FLOPs and learnable parameters with all competitors in Tab. S7. We conclude that the advantage of UniAD does not come from a larger model capacity.

## E.2    Visualization results

**Visualization results of reconstructed features** are given in Fig. S2. The feature visualization follows the approach described in Sec. C.3. MLP, CNN, and transformer all tend to learn an "identical shortcut", where the anomalous regions would also be well recovered. In contrast, our UniAD overcomes such a problem and manages to *reconstruct anomalies as normal samples*.

**Qualitative results for anomaly localization on MVTec-AD** are given in Fig. S3. All 15 categories are handled by a unified model. For both *global* (Fig. S3i-left, Fig. S3m-left) and *local* (Fig. S3b, Fig. S3h) structural anomalies, both *additional* (Fig. S3h-left, Fig. S3f-right) and *missing* (Fig. S3c-right, Fig. S3m) anomalies, both *tightly aligned objects* (Fig. S3a, Fig. S3b) and *randomly placed objects* (Fig. S3f, Fig. S3j), both *texture bumps* (Fig. S3e-left, Fig. S3g-left) and *texture scratches* (Fig. S3k-left, Fig. S3n), both *color perturbations* (Fig. S3k-right, Fig. S3h-right) and *uneven surface disturbances* (Fig. S3d, Fig. S3g), our method could successfully reconstruct anomalies to their corresponding normal samples, then accurately localize anomalous regions through reconstruction differences. This reflects the effectiveness of our UniAD.

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

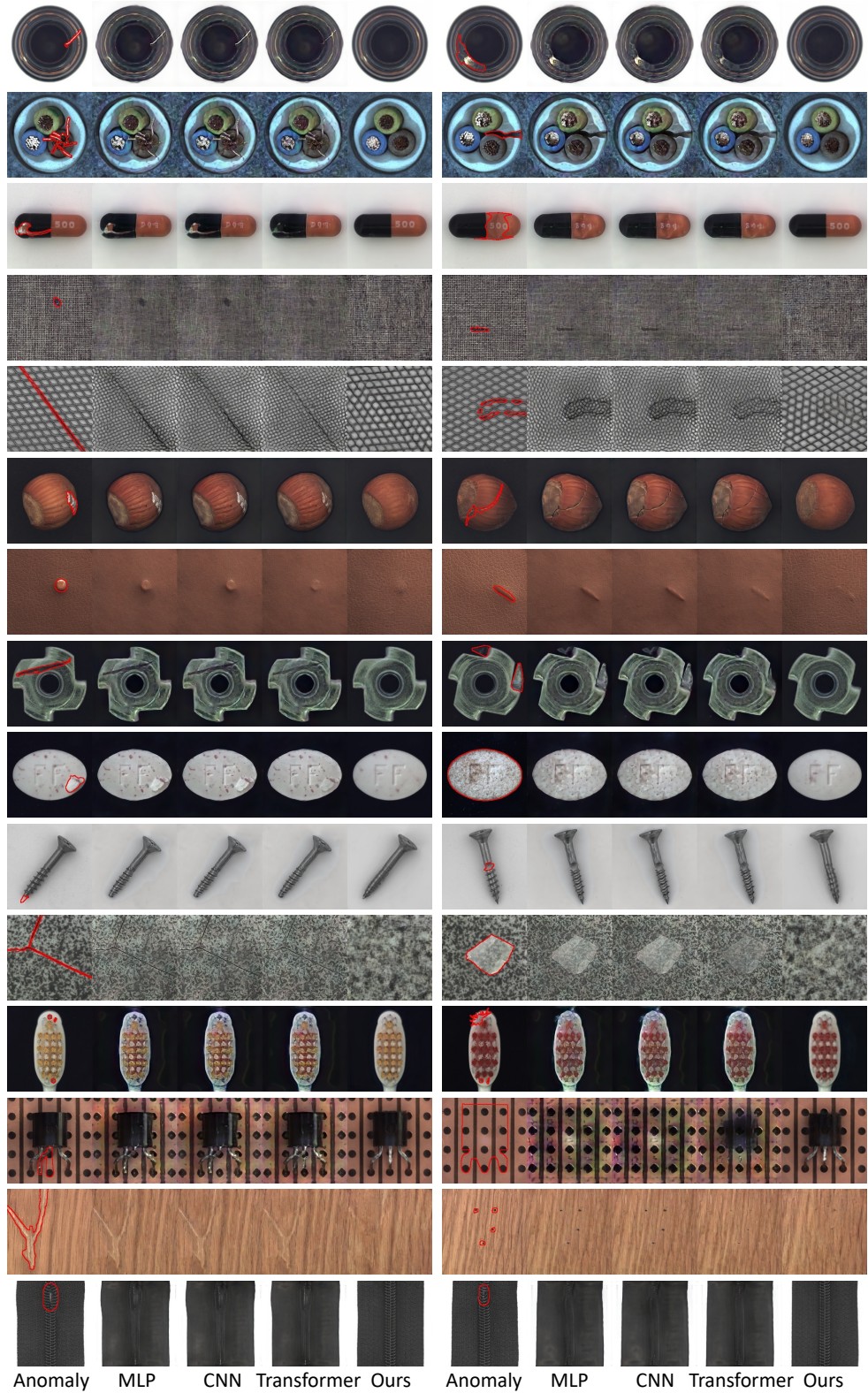

Anomaly    MLP    CNN    Transformer    Ours        Anomaly    MLP    CNN    Transformer    Ours

Figure S2: **Visual comparison** between various reconstruction approaches. MLP, CNN, and transformer all tend to learn an "identical shortcut", where the anomalous regions (highlighted by **red**) can be well recovered. In contrast, our UniAD overcomes such a problem and manages to *reconstruct anomalies as normal samples*.

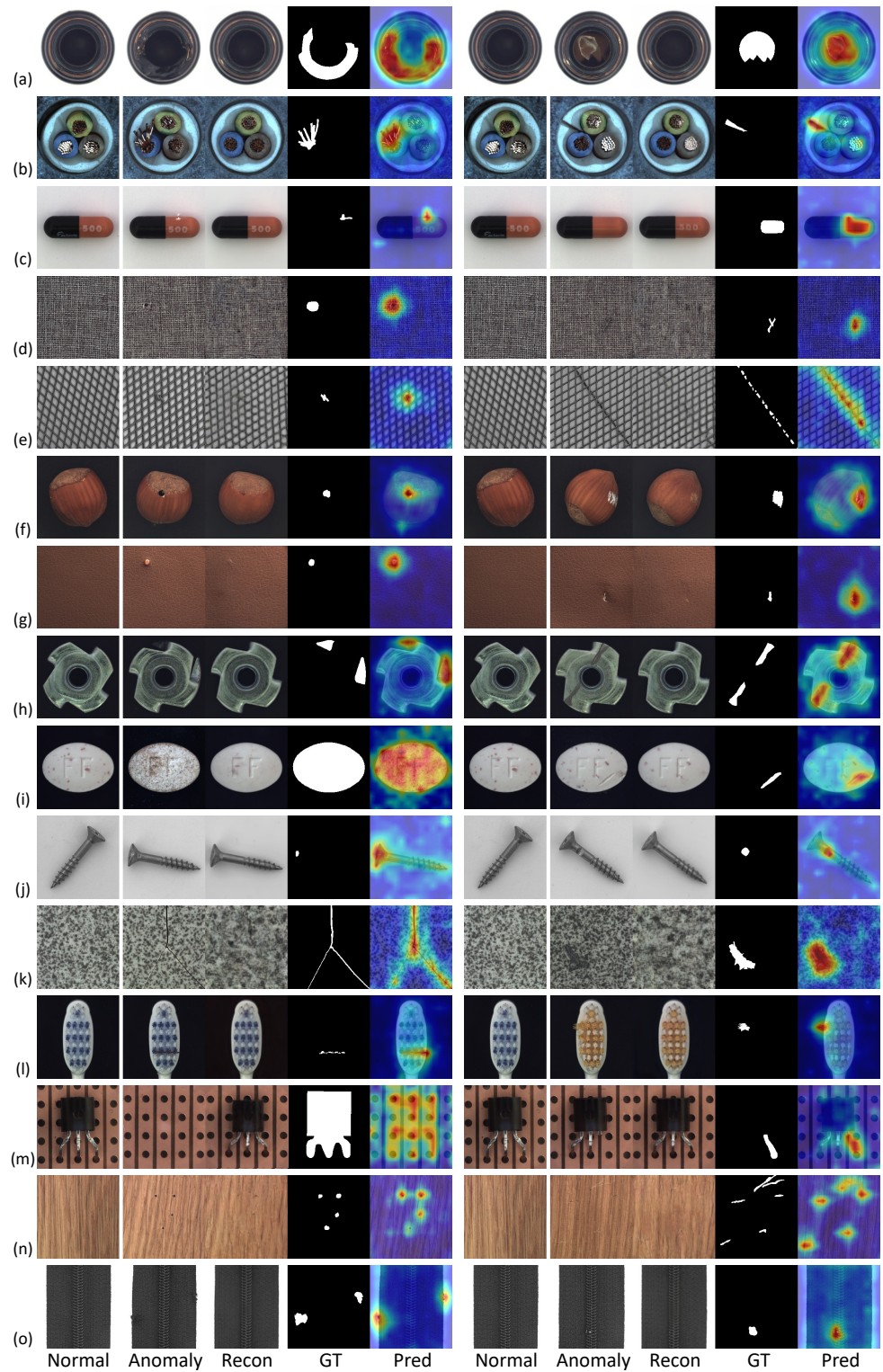

Figure S3: **Qualitative results** for anomaly localization on MVTec-AD [2] under the unified case. From left to right: normal sample as the reference, anomaly, our reconstruction, ground-truth, and our predicted anomaly map.