# OpenReview forum: "A Unified Model for Multi-class Anomaly Detection"
_NeurIPS.cc/2022/Conference — NeurIPS 2022 Accept_

### Official Review · Reviewer_SFc6 · 2022-06-29

**Rating:** 6
**Confidence:** 4
**Soundness:** 3 good
**Presentation:** 3 good
**Contribution:** 2 fair

**Summary:**

The authors propose the learning of multi-class decision boundaries for the task of anomaly detection (AD) over multiple object classes. For this, they employ reconstruction-based scores obtained from a transformer network, modified with a couple of simple tricks, such as masking neighboring points in the attention map, and increasing the capacity of the decoder. Results on MVTec show this is a promising direction for AD over multiple object classes.

**Questions:**

This paper presents strong experimental results, however appears to target a somewhat loose definition of multi-class AD. I would suggest the authors work to improve the clarity of their manuscript.

In particular:
- clarify how the proposed multi-object AD task of "unified AD" fits in or differs from existing works that assume multiple objects in the normal distribution (e.g. "semantic AD").
- clarify the novelty of the proposed transformer components, which appear highly incremental. Improvements could consist in more direct comparisons (and ablations) against transformer-based competitor models.

**Limitations:**

Limitations are discussed in Section 5, potential societal impacts (say, anomaly detection for tasks such as video surveillance) have not been discussed.

**Strengths And Weaknesses:**

# Strengths

To enable their transformer-based model to work with the task of adopting a complex normal distributions, the authors come up with some modifications, in particular neighbor-masked attention which they insert directly into the architecture to replace the default attention layer. While the authors employ other approaches such as "feature jittering", these correspond to a simple addition of Gaussian noise during the input stage, and given its simplicity I would suggest authors can remove this from abstract etc., as it doesn't present any significant novelty.

The experimental results are the strong point of this work, with outstanding performance on MVTec-AD and CIFAR10 (which is a much less relevant benchmark though, in particular as more challenging ones have recently been used, e.g. CIFAR100/STL10, c.f. works listed below).

Moreover, the paper is well-written and easy to follow.

# Weaknesses

The proposed modifications are relatively straightforward, in particular the ablations in Table 4 indicate that a vanilla transformer would end up outperforming the existing state of the art on MVTec-AD. Given there are recent works on using transformers for AD, e.g. AnoVIT (as pointed out by the authors on page 3 lines 99-101), this somewhat limits the novelty of the proposed method.

Surprisingly, from the ablation in Table 4 it appears as if feature jittering (FJ) boosts performance nearly as much as neighborhood-masked attention (NMA). I would suggest included pairwise coupling (e.g. NMA+FJ) to shed light on which ones of these are required in unison, or whether they obtain similar outcomes.

Moreover, there have been various works in the recent past that attempt to perform anomaly detection over classes that contain more than a single object class (with no labels assumed present to distinguish the objects). A discussion of these recent works, and how they relate/differ from the way in which multiple classes are treated here is missing, e.g.:
- "Deep Semi-Supervised Anomaly Detection", Ruff et al., ICLR 2020
- "Detection Semantic Anomalies", Ahmed & Courville, AAAI 2020
- "Transfer-Based Semantic Anomaly Detection", Deecke et al., ICML 2021
These manuscripts investigate the presence of multiple classes in the normal distribution. While the focus is different (latent classes), it should be compared against in this work.

# Update

The authors have addressed points raised in my original review by adding ablations and contrasting their proposed setting to existing ones. Some concerns remain around the novelty of the proposed approach.

The score has been increased to reflect the newly incorporated changes.

---

> ### Author Response · Authors · 2022-08-02
> **Response to Reviewer SFc6**
>
> **Q1: Differences from existing works involving multiple classes.**
>
> The task setting studied in this work clearly differs from "semantic AD".
>
> First, we focus more on the industrial anomaly detection dataset, MVTec-AD, which is of more practical usage. Unlike CIFAR-10, _each category has normal and anomalous samples_ in MVTec-AD. We would like to model the _joint distribution of normal samples across all categories_. It requires the model to learn "what normal samples from each category look like" instead of "what categories are normal". The latter is the main focus of "semantic AD".
>
> Second, we also differ from prior arts [1][2][3] regarding the CIFAR-10 task setting.
> * [1] studies the _one-versus-many_ setting, which treats 1 class as normal and the remaining 9 classes as anomalous.
> * [2] studies the _many-versus-one_ setting, which treats 9 classes as normal and the remaining class as anomalous.
> * [3] studies both _one-versus-many_ and _many-versus-one_ settings.
> * We study the **_many-versus-many_** setting, which treats 5 classes as normal and the other 5 classes as anomalous. We use such a setting to simulate the real scenario, where _both normal and anomalous samples contain multiple classes_.
>
> We have added the discussion of such differences in the revised version (Line 213).
>
> [1] Deep Semi-Supervised Anomaly Detection. Ruff _et al._. ICLR'20.
>
> [2] Detecting Semantic Anomalies. Ahmed and Courville. AAAI'20.
>
> [3] Transfer-Based Semantic Anomaly Detection. Deecke _et al._. ICML'21.
>
> **Q2: Comparison with transformer-based competitors.**
>
> Using transformer for anomaly detection is not our focus. We choose transformer as the reconstruction model considering its great potential in preventing the model from learning the "identical shortcut" (please refer to Sec. 3.1 in the submission). Concretely, we find that the _learnable query embedding_ is essential for avoiding such a shortcut but is seldom explored in existing transformer-based approaches [4][5][6]. As shown in the table below, after introducing even only one query embedding, our baseline already outperforms existing alternatives by a sufficiently large margin in the unified setting. Our proposed three components further improve our _strong baseline_. Recall that all three components are proposed to avoid the model from directly outputting the inputs.
>
> In short, the baseline is not a previous approach, but instead a most straightforward modification based on our revisiting of the "identical shortcut" issue. We have added the clarification and the comparison in the revised supplementary material (Sec. E2).
>
> | Method | Loc. AUROC (unified / separate) | Det. AUROC (unified / separate) | 1 query | layer-wise query |
> |  ----  | ----  | ----  | ----  | ----  |
> | InTra [4] | 70.6 / 96.6 | 65.3 / 95.0 | × | × |
> | VT-ADL [5] | 64.4 / 82.0 | 55.4 / 78.7 | × | × |
> | AnoVit [6] | 68.4 / 83 | 69.6 / 78 | × | × |
> | Ours (baseline) | 92.8 / 95.8 | 87.6 / 94.7 | ✓ | × |
> | Ours | **96.8** / 96.6 | **96.5** / 96.6 | × | ✓ |
>
> [4]  Inpainting Transformer for Anomaly Detection. Pirnay and Chai. International Conference on Image Analysis and Processing, 2022.
>
> [5] VT-ADL: A Vision Transformer Network for Image Anomaly Detection and Localization. Mishra _et al._. International Symposium on Industrial Electronics, 2021.
>
> [6] AnoViT: Unsupervised Anomaly Detection and Localization With Vision Transformer-Based Encoder-Decoder. Lee and Kang. IEEE Access, 2022.
>
> **Q3: More ablation experiments.**
>
> Please refer to **Q2 to Reviewer JpFX**. Each of the three components, _i.e._, layer-wise query embedding, neighbor masked attention (NMA), and feature jittering (FJ), could help the model not to learn the "identical shortcut" and hence bring considerable improvement on its own.
>
> **Q4: Societal impacts.**
>
> Thanks. Anomaly detection may be used for video surveillance, which may infringe personal privacy. We have updated the potential societal impacts in the revised paper (Line 313).

---

> > ### Comment · Reviewer_SFc6 · 2022-08-08
> > **Re: Response to Reviewer SFc6**
> >
> > Many thanks to the authors for their detailed response. In my original review I raised a couple of criticisms, asking for (1) clarifications w.r.t. existing AD settings, (2) novelty of the approach, and (3) additional ablations. The authors have partially addressed these. I will first reply point-by-point, and then summarize regarding (1), (2), (3) below.
> >
> > *“The task setting studied in this work clearly differs from "semantic AD”.”*
> >
> > I fully agree, which is why I wanted to encourage a detailed discussion in the manuscript.
> >
> > *“We have added the discussion of such differences in the revised version (Line 213).”*
> >
> > Thank you for adding this discussion. Given the many recent submissions that proposed new AD settings, I believe a careful explanation of their differences (in particular with regards to the AD problem introduced here) enhances the paper.
> >
> > *“Using transformer for anomaly detection is not our focus. We choose transformer as the reconstruction model considering its great potential in preventing the model from learning the "identical shortcut" (please refer to Sec. 3.1 in the submission)”*
> >
> > From the abstract and instruction readers will in all likelihood get a different impression, as considerable focus is put on the different elements used in the transformer-based architecture; for example “we propose a feature jittering strategy” [L13] is highlighted in the abstract. To readers, this raises the curiosity what methods will be introduced throughout — unfortunately I maintain that the actual methodologies are a little underwhelming in terms of their technical novelty, given the space awarded to them in the manuscript.
> >
> > After reading other reviewer’s concerns and respective author’s comments, my concerns w.r.t. ablations have been accounted for, thank you for adding these.
> >
> > ## Summary
> >
> > The authors have addressed (1) and (3), some concerns regarding (2) novelty of the methods in the manuscript remain — please note the score has been increased.

---

> > > ### Author Response · Authors · 2022-08-08
> > > **Thanks.**
> > >
> > > Thank you for your valuable suggestions that help us improve the manuscript. We are glad that you appreciate the "identical short" problem studied in this work, which is our major focus. In the meantime, we also agree that our current presentation (*i.e.*, abstract and introduction) may give too much space to the solutions in solving the "identical short" issue, which slightly upstage the problem itself. As suggested, we will rephrase some sentences to highlight the *novel problem setting* as well as the *reasons on why it is challenging* in the next version, such that the readers can have a better understanding of the scope of this work.
> > >
> > > Thank you again for your effort in the review and the discussion!

---

> ### Author Response · Authors · 2022-08-08
> **Any more comments or concerns?**
>
> Dear reviewers and AC
>
> Thanks a lot for your effort in reviewing this submission! We have tried our best to address the mentioned concerns/problems in the rebuttal. Feel free to let us know if there is anything unclear or so. We are happy to clarify them.
>
> Best,
> Authors

---

### Official Review · Reviewer_JpFX · 2022-07-07

**Rating:** 5
**Confidence:** 5
**Soundness:** 3 good
**Presentation:** 3 good
**Contribution:** 2 fair

**Summary:**

This paper aims to learn a unified framework for detecting multi-class anomalies. Anomaly detection is only trained on normal data, so the so-called “identical shortcut” phenomenon may occur. To solve this problem, this paper proposes the following strategies: 1) layer-wise query decoder, 2) neighbor masked attention, 3) feature jittering. The experimental results on the MVTec-AD and CIFAR-10 datasets show that the proposed method can alleviate the “identical shortcut” phenomenon.


**Questions:**

1. On lines 125-126, the author mentions that the model may learn a trivial solution, causing anomaly detection to fail. But in MLP, this should not be the case due to the presence of nonlinear activations. Can the authors explain this further through experiments or visualizations?
2. In Table 2, for the categories Capsule and Transistor, the results of the method DRAEM (50.5 and 64.5) are significantly smaller than those of the method in this paper (98.5 and 97.9), but for the category Zipper, the results of the method DRAEM (98.3) are higher than those of the method in this paper (96.8). What are the reasons for the above phenomenon? Please analyze this.
3. Please analyze the complexity of the method in this paper compared with the one-class-one model method.


**Ethics Review Area:**

["I don’t know"]

**Limitations:**

The authors have adequately addressed the limitations and potential negative societal impact of their work.

**Strengths And Weaknesses:**

Strengths
1. This paper proposes a novel neighbor masked attention.
2. This paper is well organized, easy to understand, and clearly written.
3. Good results are achieved on the MVTec-AD and CIFAR-10 datasets.

Weaknesses
1. The introduction to evaluation metric (AUROC) is missing.
2. The ablation experiments in Table 4 are incomplete. For example, under 1 q., the results in the case of only NMA and FJ are missing; under Layer-wise q., the results in the case of only NMA are missing; under Layer-wise q., the results in the case of only FJ are missing.
3. Feature jittering is a common practice and lacks novelty.

---

> ### Author Response · Authors · 2022-08-02
> **Response to Reviewer JpFX**
>
> **Q1: Introduction to evaluation metric (AUROC).**
>
> Area Under the Receiver Operating Curve (AUROC) follows the standard evaluation protocol for the MVTec-AD dataset. It is independent of the threshold used to detect anomalies. To obtain the receiver operating curve, the true positive rate is defined as the percentage of pixels (for anomaly localization) or images (for anomaly detection) that are accurately identified as anomalies. The false positive rate is defined as the percentage of pixels or images that were wrongly classified as anomalies. We have included the introduction of AUROC in the revised version (Line 216).
>
> **Q2: More ablation experiments.**
>
> Thanks. We provide the full ablation experiments regarding layer-wise query embedding, neighbor masked attention (NMA), and feature jittering (FJ) as below. We can tell that, all these three components could boost the performance, namely layer-wise query with 3.7\% gain (92.8\% to 96.5\%), NMA with 3.5\% gain (from 92.8\% to 96.3\%), and FJ with 3.0\% gain (from 92.8\% to 95.8\%). It demonstrates that all these designs can help the model _not_ to learn the "identical shortcut". Furthermore, combining all three components brings the best performance. This table is also included in Tab. S2 of the revised supplementary material (Page 4).
>
> | w/o query | 1 query | layer-wise query | NMA | FJ | Loc. AUROC | Det. AUROC |
> |  ----  | ----  | ----  | ----  | ----  | ----  | ----  |
> | ✓ | - | - | - | - | 79.4 | 69.5 |
> | - | ✓ | - | - | - | 92.8 | 87.6 |
> | - | ✓ | - | ✓ | - | 96.3 | 96.1 |
> | - | ✓ | - | - | ✓ | 95.8 | 95.0 |
> | - | ✓ | - | ✓ | ✓ | 96.6 | 96.2 |
> | - | - | ✓ | - | - | 96.5 | 95.0 |
> | - | - | ✓ | ✓ | - | 96.5 | 95.8 |
> | - | - | ✓ | - | ✓ | 96.2 | 94.9 |
> | - | - | ✓ | ✓ | ✓ | **96.8** | **96.5** |
>
> **Q3: Feature jittering is a common practice and lacks novelty.**
>
> Although feature jittering (FJ) is simple, it _rightly fits our motivation_, which is to prevent the model from learning the "identical shortcut". Concretely, it urges the model to recover the correct information from noisy inputs, leaving the "identical shortcut" un-optimal anymore. In other words, the model is forced to learn semantic knowledge instead of directly outputting the inputs. The ablation studies in **Q2** also verify the effectiveness of FJ.
>
> Our major contributions lie in (1) defining a more challenging task setting for anomaly detection, (2) analyzing how the "identical shortcut" harms the performance of anomaly detection, and (3) proposing three feasible solutions. We believe this study could inspire more explorations along this direction.
>
> **Q4: Activations in MLP.**
>
> Thanks. The revisiting in Sec. 3.1 is a _rough_ analysis of the "identity shortcut" problem. Providing a rigorous analysis could be far more challenging and not under the main scope of this work. But in our experiments shown in Fig. 2, all MPLs, CNNs, and Transformers have non-linear activations. We can still observe the phenomenon that "loss gets smaller yet the performance drops". This empirically verifies our claim. We have revised the paper (Line 121) to tell the readers that the analysis is not rigorous.
>
> **Q5: Analysis of DRAEM.**
>
> DRAEM relies on some simulated anomalies generated with Perlin noise. These pseudo-anomalies are very alike the actual anomalies of some categories, like Zipper and 5 Texture categories (_e.g._, usually present color perturbations or texture discontinuities). However, for some categories, like Capsule, Metal Nut, and Transistor (_e.g._, usually present structural anomalies), the pseudo-anomalies are clearly different from those real anomalies. Therefore, in the unified case, all these categories are trained together with the same simulation strategy, making the model prone to easy-to-learn ones.
>
> **Q6: Complexity comparison.**
>
> With the image size fixed as $224 \times 224$, we compare our UniAD with all competitors regarding the inference FLOPs and learnable parameters in the table below. We can tell that the advantage of our approach does not come from a larger model capacity. This table is also included in Tab. S7 of the revised supplementary material (Page 6).
>
> | | US | PSVDD | PaDiM | CutPaste | FCDD | MKD | DRAEM | Ours |
> |  ----  | ----  | ----  | ----  | ----  | ----  | ----  | ----  | ----  |
> | FLOPs (G) | 60.32 | 149.74 | 23.25 | 3.65 | 13.16 | 32.11 | 245.15 | 6.46 |
> | Learnable Params (M) | 9.55 | 0.41 | 950.36 | 13.61 | 4.51 | 0.34 | 69.05 | 7.48 |

---

> > ### Comment · Reviewer_JpFX · 2022-08-08
> > **Reply to Response**
> >
> > Thanks to the author for the reply. Most of my concerns were addressed. Anomaly detection in images is a very interesting topic. I hope the author releases the code and bring progress to the community.

---

> > > ### Author Response · Authors · 2022-08-08
> > > **Thanks.**
> > >
> > > Thanks for liking our work. We will release the code.

---

> ### Author Response · Authors · 2022-08-08
> **Any more comments or concerns?**
>
> Dear reviewers and AC
>
> Thanks a lot for your effort in reviewing this submission! We have tried our best to address the mentioned concerns/problems in the rebuttal. Feel free to let us know if there is anything unclear or so. We are happy to clarify them.
>
> Best,
> Authors

---

### Official Review · Reviewer_npjB · 2022-07-17

**Rating:** 6
**Confidence:** 3
**Soundness:** 3 good
**Presentation:** 3 good
**Contribution:** 4 excellent

**Summary:**

The paper tackles anomaly detection of multiple classes without class labels. In other words, the proposed method learns the normality of multiple classes at once without the need for class label information. The paper analyzes that the reconstruction-based anomaly detectors learn ‘identity shortcut’ and introduces techniques how to prevent this phenomenon. To this end, the paper proposes three techniques: the use of query embedding in multiple layers of a transformer, neighbor masked attention and feature jittering. The experiments are conducted on MV-Tech and CIFAR10 datasets.

**Questions:**

How did the proposed method and competing methods are trained for the unified and separate case? In the separate case, are they trained on the whole dataset and finetuned for each class-wise dataset? It is unclear how the models are trained in each scenario.

**Limitations:**

Limitations are addressed in Section5.

**Strengths And Weaknesses:**

## Strength

### Problem setup
Anomaly detection on a multi-mode dataset (multi-class dataset without class information) is a relatively underexplored area. Most of the out-of-distribution detection papers assume class information given. In addition, this paper targets anomaly localization tasks.

### Analysis and extensive ablation study supports the idea
- Section 3.1 shows the performance deprecation over the training epochs showing that reconstruction-based models’ performance is unstable during training (the phenomenon of identical shortcuts).
- Section4.5 includes extensive ablation studies supporting the design of the method and sensitivity in each component and hyperparameters. Most hyperparameters are insensitive to performance showing less than ~1% gap.

### Strong performance
The proposed method shows a notable performance gap over competing methods in a unified scenario on MV-tech and CIFAR 10.


## Weakness

### Lack of analysis

- The design of the method is not targeted for a unified case but is effective. Why?
The idea of the proposed method uses general ML techniques not tailored for unified (multi-class data without label information) cases, yet effective. What would be the main reason for this? How is the problem of learning identical-shortcut relevant to the performance of unified anomaly detection scenarios?

- Why does the proposed method not gain performance when label information is added? (Separate case) In Table1 and Table2, the proposed method does not improve much by adding label information.

---

> ### Author Response · Authors · 2022-08-02
> **Response to Reviewer npjB**
>
> **Q1: Relation between the "identical shortcut" problem and the unified case.**
>
> Thanks. The "identical shortcut" issue is a general problem in auto-encoder networks. As a result, all reconstruction-based anomaly detection methods would face this risk. However, under the unified case, where the distribution of normal data is more complex, the "identical shortcut" problem is magnified. Intuitively, to learn a unified model that can reconstruct all kinds of objects, it requires the model to work extremely hard to learn the joint distribution. From this perspective, learning an "identical shortcut" appears as a far easier solution.
>
> In Fig. 2a of the submission, we aim to visualize the "identical shortcut" issue, where the loss becomes smaller yet the performance drops. We conduct the same experiment under the separate case. As shown in Fig. S1 of the revised supplementary material (Page 2), the accuracy keeps growing up along with the loss getting smaller. This helps reveal the relation between the "identical shortcut" problem and the unified case, which is that _the unified case is more challenging and hence magnifies the "identical shortcut" problem_. Therefore, although our approach is not specially designed for the unified case, such a challenging task clearly highlights our strengths over existing alternatives.
>
> **Q2: Performance gain from the unified case to the separate case.**
>
> We do not use "label" information for the separate case. For the separate case, we train 15 separate models, each for a single category, following prior arts. Consequently, the separate models are learned _exactly the same_ as the unified model, but _on some easier data distributions_. From this perspective, our approach does _not_ suffer from a performance drop when the to-learn distribution gets more complex. By contrast, existing alternatives only perform well on simple distributions yet fail to handle such a challenging task. Thus, instead of saying there is no performance gain from the unified case to the separate case, what we want to express is that _there is no performance drop from the separate case to the unified case_.
>
> **Q3: Training setup.**
>
> For both the unified case and the separate case, the model learns on a collection of unlabeled images from scratch. The only difference is that whether the image collection comes from one object category or multiple categories. We have clarified this in the revised version (Line 227).

---

> > ### Comment · Reviewer_npjB · 2022-08-03
> > **I suggest to add the details of experiment S1 into the main paper.**
> >
> > Thank you for the feedback.
> >
> > My concerns are addressed in the revised version of the paper.
> > The authors show that 'The unified case is more challenging and hence magnifies the identical shortcut problem' with the additional experiment in S1. I think this is an important message of the paper and should be in the main text with detailed discussions.
> > Without a clear demonstration of the difference in identical shortcut problem between unified and separate anomaly detection settings, the contributions claimed in this paper are less plausible.

---

> > > ### Author Response · Authors · 2022-08-04
> > > **We will merge experiment S1 into the main paper.**
> > >
> > > Thanks for your suggestion. We have already planned to merge some necessary materials in the supplementary material into the main paper. Currently, we leave the revised version and the supplementary material as their current form to help the reviewers and ACs *track the revision* (*i.e.*, newly added materials).

---

> ### Author Response · Authors · 2022-08-08
> **Any more comments or concerns?**
>
> Dear reviewers and AC
>
> Thanks a lot for your effort in reviewing this submission! We have tried our best to address the mentioned concerns/problems in the rebuttal. Feel free to let us know if there is anything unclear or so. We are happy to clarify them.
>
> Best,
> Authors

---

### Meta-Review · Area_Chair_ieg9 · 2022-08-26

**Recommendation:** Accept
**Confidence:** Certain

**Metareview:**

This paper is on a highly-important topic, and makes solid contributions. Anomaly detection for multi-class datasets without class information is an underexplored area. Reviewers have appreciated the strong experimental results (especially on the important MVtech benchmark), high quality paper writing, and explainability results besides accuracy, via a novel attention mechanism. On the flip side, there were concerns on lack of deep analyses of the constituents of the method and novelty (given that there are some recent papers with similar ideas). The scores were borderline and the authors have put significant effort to address the concerns of the reviewers. Especially extra ablation studies and comparisons with other relevant papers are quite helpful in regards to convincingness of the ideas. I support the acceptance of the paper given all. Please update your paper with the additional content you have provided in the responses below.

**Award:**

No

---

### Decision · Program_Chairs · 2022-09-14

Accept